# GOTRIPLE: A User-Centric Process to Develop a Discovery Platform

**Suzanne Dumouchel [1], Emilie Blotière [1,\*], Gert Breitfuss [2], Yin Chen [3], Francesca Di Donato [4], Maria Eskevich [5], Paula Forbes [6], Haris Georgiadis [7], Arnaud Gingold [8], Elisa Gorgaini [5], Yoann Moranville [9], Stefanie Pohle [10], Stefano de Paoli [6], Clara Petitfils [1] and Erzsebet Toth-Czifra [9]**

[1] CNRS (Huma-Num), 75016 Paris, France; suzanne.dumouchel@huma-num.fr (S.D.); clara.petitfils@huma-num.fr (C.P.)

[2] Know-Center, 8010 Graz, Austria; gbreitfuss@know-center.at

[3] EGI, 1098 XG Amsterdam, The Netherlands; yin.chen@egi.eu

[4] Consiglio Nazionale delle Ricerche (CNR-ILC), 56124 Pisa, Italy; francesca.didonato@ilc.cnr.it

[5] CLARIN, 3512 BS Utrecht, The Netherlands; maria@clarin.eu (M.E.); e.gorgaini@uu.nl (E.G.)

[6] School of Business, Law and Social Sciences, Abertay University, Dundee DD1 1HG, UK; p.forbes@abertay.ac.uk (P.F.); s.depaoli@abertay.ac.uk (S.d.P.)

[7] National Documentation Centre (EKT), Athina 116 35, Greece; hgeorgiadis@ekt.gr

[8] CNRS (OpenEdition), 75016 Paris, France; arnaud.gingold@openedition.org

[9] DARIAH, D-10117 Berlin, Germany; yoann.moranville@dariah.eu (Y.M.); erzsebet.toth-czifra@dariah.eu (E.T.-C.)

[10] Max Weber Stiftung (MWS), D-53173 Bonn, Germany; pohle@maxweberstiftung.de

\* Correspondence: emilie.blotiere@huma-num.fr

**Abstract:** Social sciences and humanities (SSH) research is divided across a wide array of disciplines, sub-disciplines and languages. While this specialization makes it possible to investigate the extensive variety of SSH topics, it also leads to a fragmentation that prevents SSH research from reaching its full potential. The TRIPLE project brings answers to these issues by developing an innovative discovery platform for SSH data, researchers' projects and profiles. Having started in October 2019, the project has already three main achievements that are presented in this paper: (1) the definition of main features of the GOTRIPLE platform; (2) its interoperability; (3) its multilingual, multicultural and interdisciplinary vocation. These results have been achieved thanks to different methodologies such as a co-design process, market analysis and benchmarking, monitoring and co-building. These preliminary results highlight the need for respecting diversity of practices and communities through coordination and harmonization.

**Keywords:** user-centric approach; user research; social sciences and humanities; open science; European Open Science Cloud (EOSC); FAIR principles; discovery; research data

## 1. Introduction

Open data are an open window to the world, accessible to the greatest number of users. Retrieving information and knowledge comes with significant challenges when trying to avoid transforming this opportunity into a disorganized and indigestible mass of data through a scattergun approach. This is why, in parallel with the technological challenges, we are particularly attentive to the needs of users as varied as a scholar or scientist, company director, policy maker, student or simply a citizen who does research for his or her own pleasure. The aim of the GOTRIPLE platform (developed by the TRIPLE project, https://www.gotriple.eu/) is to make it much easier for scientists,

citizens and business organizations to access scientific publications, data, data processing platforms and data processing services and therefore to benefit from Open Science. Open Science represents a new approach to the scientific process based on cooperative work and new ways of diffusing knowledge by using digital technologies and new collaborative tools [1] (p. 33). The OECD (Organisation for Economic Co-operation and Development) defines Open Science as: "to make the primary outputs of publicly funded research results—publications and the research data—publicly accessible in digital format with no or minimal restriction" [2] (p. 7), and they add another important aspect to the concept: "Open Science is about extending the principles of openness to the whole research cycle, fostering sharing and collaboration as early as possible thus entailing a systemic change to the way science and research is done." [3].

Through the user-centered approach which characterizes the TRIPLE project, the discovery platform aims at fostering the creation, development and strengthening of the layer of researchers in the social sciences and humanities (SSH) both in Europe and worldwide. The "Community of Practice" [4] concept is our basis for conducting research. The concept will be harnessed to capture the idea that a group of people who have a common interest in a certain area can deliver better learning and improved results by working together and sharing expertise, which benefits the larger collective. Through its multilingual and multicultural discovery tool GOTRIPLE, the TRIPLE project brings together members of the scientific community from different fields, languages, countries and communities in research projects to ensure that they collaboratively will be capable of offering improved solutions to research problems. Indeed, by more easily identifying the skills of researchers, the discovery platform has been designed to foster new collaborations and exchanges among members of the scientific community.

The platform, for which development process is on-going, will provide linked exploration thanks to aggregators such as (1) the ISIDORE search engine (a large-scale discovery service, developed by Huma-Num since 2009 (https://isidore.science/) and (2) a variety of connected innovative tools, which include visualizations, a web annotation service, a trust building system, a crowdfunding tool and a recommender system. Through a user-centric approach and a set of methodologies that will be described in Section 3, our main objective is to enable researchers to discover and reuse SSH data macro-typologies, related not only to publications, but also to people (researchers) and projects. The TRIPLE solution supports Open Science principles, especially Open Access and Findable, Accessible, Interoperable and Re-usable (FAIR) data.

## 2. Results

Although TRIPLE is still at the early stage of the development, thanks to a consortium of experts in their fields (researchers, data engineers, and staff from small and medium enterprises in the field of Information and Communications Technology), the project has already achieved some tangible results, communicated via deliverables and other working papers. Since the platform will not be completed until 2023, the results presented here are preliminary. The first tasks have already yielded results confirming the need for a new discovery platform dedicated to Social Sciences and Humanities. Described here are the three preliminary results:

Result 1: Definition of the main features of the GOTRIPLE platform: It aims at meeting the needs of researchers and other stakeholders by allowing researchers to make their way through millions of documents and bring together members of the scientific community from different fields, countries and communities in research projects to foster collaboration across the frontiers of countries and disciplines and increase the impact of research in societal issues. A discovery service is the core of the platform and highlights the skills and competences of researchers, to encourage efficient collaboration according to the needs of researchers. However, various innovative services will be plugged into the platform allowing researchers to share annotated documents and to envisage interdisciplinary collaborations via networking services based on trust and recommendation. A crowdfunding service is also planned to foster bridges between research and societies, to make research accessible to a wide range of people and to encourage the impact of SSH discoveries in civil society.

Result 2: Interoperability of the GOTRIPLE platform, especially regarding the European Open Science Cloud. This result can be seen through two achievements: The platform is compliant with the FAIR principles (https://www.go-fair.org/fair-principles/).

The first of the FAIR principles, Findability, is at the heart of the building of a discovery platform. Technically, findability is supported by the use of Persistent Identifiers (PIDs), either harvested from the providers' repositories or generated by the platform, for each searchable element; PIDs are registered in the metadata record, while rich minimal metadata facilitate data discovery thanks to the establishment of a TRIPLE model using schema.org. Concerning Accessibility, while all the previous findability features will be part of the search interface, data and metadata will be also accessible through free, open and documented protocols, namely: OAI-PMH (Open Archives Initiative Protocol for Metadata Harvesting), SPARQL (Protocol and RDF Query Language) endpoint, and APIs (Application Programming Interface). Concerning interoperability, GOTRIPLE will tag variable-level information in the most relevant open standards for SSH i.e., in the Data Documentation Initiative (DDI), Text Encoding Initiative (TEI), Metadata Encoding and Transmission Standard (METS), and Metadata Object Description Schema (MODS). Metadata records produced by GOTRIPLE will be published using the following standard vocabularies: Component MetaData Infrastructure, Dublin Core Metadata Element Set and DCMI (Dublin Core Metadata Initiative) Metadata Terms. Moreover, metadata records published in RDF (Resource Description Framework) will use the following linked open data vocabularies: the Data Catalog Vocabulary (DCAT), Open Digital Rights Language (ODRL), DDI-RDF Discovery Vocabulary (Disco). Lastly, TRIPLE will ensure the reusability of all the content that the project will create: the project grants Open Access to all project results, which will be published in Open Access journals (Gold road) and, when relevant, deposited in Open Access repositories (Green road). All data and metadata (with the exclusion of the user research data) will be available in Open Access with open licenses allowing reuse. Furthermore, TRIPLE is working closely with the data providers in order to have a consistent licensing policy both for data and for metadata.

The authentication portal is compliant with the other OPERAS services, with EGI services (as it relies on EGI check-in) and with the European Open Science Cloud (EOSC) AAI: Such EOSC Federation Services include, but are not limited to, the Authentication and Authorization Infrastructure Authorization (AAI), the Helpdesk, the Accounting Service and the Monitoring Service. Some, such as the Accounting Service, probably have little to do with TRIPLE's aim, but others, especially the Helpdesk or the AAI can be important additions to our platform. For instance there are three different levels of integration of the Helpdesk with external services, where TRIPLE could be positioned: 1. Direct usage of the EOSC Helpdesk by the TRIPLE team (answers and follow-up happen on the EOSC Helpdesk), 2. A Ticket redirection from the EOSC Helpdesk towards the TRIPLE Helpdesk (or other Helpdesk) is performed, likely via an automatic email notification, and 3. Full integration thanks to the use of OTRS APIs between EOSC Helpdesk and the Service Helpdesk where the issue is then taken care of. However, the EOSC Helpdesk would only be available from 2021 at best (possibly even after 2023), after testing and validation is done by the various stakeholders.

Result 3: A multilingual, multidisciplinary and multicultural platform. GOTRIPLE brings together members of the scientific community from different fields, countries and communities in research projects and ensures that they collaboratively will be capable of offering improved solutions to research problems. Indeed, by more easily identifying the skills and competences of researchers in the social sciences and humanities (SSH) in Europe, GOTRIPLE will foster new collaborations and exchanges among members of the scientific community, i.e., nearly 450,000 SSH researchers in Europe. Connections will be multidirectional as in a network, alongside multiple scientific and multilingual thesaurus, by tapping into the power of LoD5 (Lines of Development) provided by Wikidata's huge corpus and through the power of social networking. That is how TRIPLE will help to create, develop and strengthen communities of SSH researchers both in Europe and worldwide. It will offer a way to citizens to experiment with a qualitative linguistic, cultural and disciplinary diversity through the discovery solution. Specialized on social sciences and humanities, TRIPLE deals mainly with cultural

and social practices in the European societies and helps them to better understand their assets and challenges in terms of identity. It will contribute to the promotion of cultural diversity inside Europe.

## 3. Methods

The TRIPLE consortium, composed of 19 partners with different expertise and competences, with complementary skills and with different approaches, working together towards a common objective, is the community that drives the design and development of the GOTRIPLE platform. The different methods adopted within TRIPLE together lead together to the implementation of GOTRIPLE. This variety of methods depends both on the different disciplinary approaches required, and on the variety of materials that have to be exploited at several steps and levels of the project. At some point, some of these materials can be reused in a different manner to achieve or contribute to another objective.

### 3.1. Methods for Result 1: Definition of the Main Features of the GOTRIPLE Platform

To define the main features of the GOTRIPLE platform, two complementary methods have been used: (1) a co-design process (user-centric approach) and (2) market analysis and benchmarking of similar or competitive platforms. By doing so, the users' perspective and the service providers' perspective have jointly contributed to define the main features of GOTRIPLE.

### 3.1.1. A User-Centric Approach

It is paramount to the relevance of a project like TRIPLE to obtain a deep and qualitative understanding of the end users and to involve them in taking relevant decisions about how the platform and its associated services can support their research goals and activities. A user-centered perspective [5] has been adopted for the design of TRIPLE. This involves working in close contact with end-users, both researchers and other stakeholders, and to investigate their specific needs regarding a discovery platform.

For the initial identification of the needs of end-users, in order to prepare the ground for targeted co-design and to support an initial definition of the platform requirements, we conducted a number of qualitative end-user interviews and developed and distributed an Europe-wide questionnaire. The qualitative interview is a research tool which has been the basis for many important studies across a range of disciplinary fields in the social sciences [6] but also in Information Systems Design [7]. With qualitative interviewing it is possible to explore people's understandings of their lives (e.g., their work, their aspirations etc.) and also many aspects of their life-long professional experiences (e.g., collaborations with colleagues). Two sets of qualitative interview scripts were prepared to explore end-user needs for the platform. The first script concentrated on investigating the needs of researchers from the social sciences and humanities (SSH) and the second one the needs of other stakeholders (e.g., public administrations, owners of SMEs, policy-makers). In addition, a questionnaire was conducted, aimed at SSH researchers, with the purpose of mapping their existing practices and services and to obtain a broader overview of their needs.

Interviews were analyzed with an inductive methodology, in particular thematic analysis [8], an approach which focuses on identifying recurring patterns and points of interest in the data. The identification of patterns is fundamental for the identification of needs and commonalities across SSH practices and across the variety of people being interviewed. With the data analyzed and a set of patterns identified, the next step of the methodology was that of building a set of project "personas" (a set of user archetypes representing relevant patterns from the interviews) and developing usage "scenarios" (narratives/stories of the personas using the platform) [9]. Questionnaire data is currently being analyzed with descriptive statistics in order to identify any differences between the demographics and to explore, on a higher level, the end-user needs.

The identification of end-users' needs offers fundamental material for conducting co-design activities for the next phase of the user-centered research. The project will draw upon the approach of

participatory design (PD) [10], especially focusing attention on any differences, for example, between disciplines or career levels. In simple terms, this is a process whereby users work directly with the designers in designing the technologies or products they will use. The most common way of conducting PD is through collaborative workshops aimed at generating shared solutions to specific problems, such as the co-design of specific innovative services for TRIPLE (e.g., a recommender system). Following the interviewing phase, which concluded in May 2020, and the questionnaire analysis (in progress, concluding in November 2020) a series of co-design workshops will be run with SSH researchers and other key stakeholders. These sessions will be supported by the personas/scenarios developed from the qualitative user research, as well as any early platform design concepts created from the user needs. In this way, the design of the platform can develop in an iterative manner, with early ideas being rediscussed during later workshops to gather further inputs from the end-users into how well the solutions meet their needs. Some workshops with single stakeholder groups will be run whilst others will include multiple stakeholders in the same session. Both traditional paper-based methodologies as well as more innovative technological approaches were planned to be used. However, due to the new social distancing imposed by COVID-19 restrictions, all the workshops will be conducted online with the use of virtual whiteboarding tools (such as Miro, https://miro.com/ and/or Mural, https://www.mural.co/).

### 3.1.2. Market Analysis and Benchmark Activities

For a service product to be successful, it not only depends on the quality of its design and development, but also on market demands and success in competition. TRIPLE carefully studies the context in which the platform is developed: from the point of view of user requirements, but also from a competitive vantage point. To gain a deep insight into the already existing offers, we carried out an extensive competitor analysis. This analysis allows us to identify and understand competitors' strengths and weaknesses in relation to the service developed by TRIPLE and helps us to develop effective competitive strategies. The results are described in detail in deliverable D7.1 "Report on Stakeholder and Opportunity Analysis" [11].

A list of competitor platforms that offer similar services and share target markets was created in collaboration with the project members. A total of forty-seven platforms were identified as potential challengers. The Alexa rank score (www.alexa.com) was used to determine these platforms' popularity. It is a global ranking system which considers the estimated average of daily unique visitors and the number of page views over the past three months. The top ten ranked platforms were included in the competitors' analysis. In order to represent the competitive environment of the TRIPLE platform in the best possible way concerning different platform types and geographical origins, a further 16 platforms were included in the analysis. Table 1 lists the 26 competitor platforms ordered by popularity.

To gain a good understanding of the competitive environment of the TRIPLE platform, information about the 26 platforms were retrieved from their websites and documented in a template. The essence of this documentation was then transferred to a summary table and analyzed through qualitative content analysis, with a focus on offered platform features and functions, organizational insights, strengths and weaknesses as well as insights into usability and user experience. To complement the vantage points gained from the web-based competitor analysis, an interview study with general Open Science experts (3 participants) and executives from existing scholarly communication platforms (6 participants) was conducted. The qualitative interviews were designed as guideline-based expert interviews and evaluated through qualitative content analysis.

Since TRIPLE's discovery tool aims at enabling users to find Open Access research data in the social sciences and humanities (SSH), we additionally analyzed the competitors with respect to access modalities and open content. The 26 competitor platforms were classified as "open-access", "partly open-access" or "non-applicable". We considered 14 platforms to be open-access because the research outputs are freely available online, and because there are no access barriers, e.g., copyright and licensing restriction or premium accounts. Nine platforms were considered partly open-access

because some access barriers were present, even though part of the content is open-access. For three platforms this classification could not be applied. Figure 1 shows the assigned platforms according to access modalities and open content.

**Table 1.** Overview of analyzed platforms (Note: The Alexa site rank uses the root address of the platform, therefore the score for Google Scholar and Elsevier Data Search refer to Google and Elsevier).

| Competitor Platform | Type of Platform | Alexa Site Rank |
| --- | --- | --- |
| Google Scholar | academic search engine | 1 |
| Researchgate | science-oriented social media | 165 |
| Academia.edu | science-oriented social media | 238 |
| Elsevier Data Search | academic search engines | 6292 |
| Semantic Scholar | academic search engines | 1124 |
| JSTOR | search engines and directories for OA (Open Access) resources | 1247 |
| arXiv.org | search engines and directories for OA resources | 2129 |
| Frontiers | academic search engines | 3641 |
| Mendeley | science-oriented social media | 4169 |
| ORCID | multidisciplinary academic databases | 5151 |
| CORE | search engines and directories for OA resources | 5660 |
| Zotero | science-oriented social media | 13,117 |
| zenodo | Repositories—institutional or subject | 50,563 |
| Center of Open Science | dissemination platform | 58,660 |
| Nextstrain | disciplinary academic database | 62,172 |
| figshare | repositories—institutional or subject | 71,192 |
| ScienceOpen | dissemination platform | 212,714 |
| unpaywall | search engines and directories for OA resources | 220,255 |
| Lens.org | academic search engines | 311,403 |
| OpenAIRE Explore | repositories—institutional or subject | 369,908 |
| Humanities commons | science-oriented social media | 383,307 |
| DataCite | multidisciplinary academic databases | 407,533 |
| Iris.ai | academic search engines | 619,629 |
| Isidore | academic search engines | 1,523,750 |
| Biblissima | shadow library | 4,535,602 |
| huni | library catalogues and discovery systems | n.a |

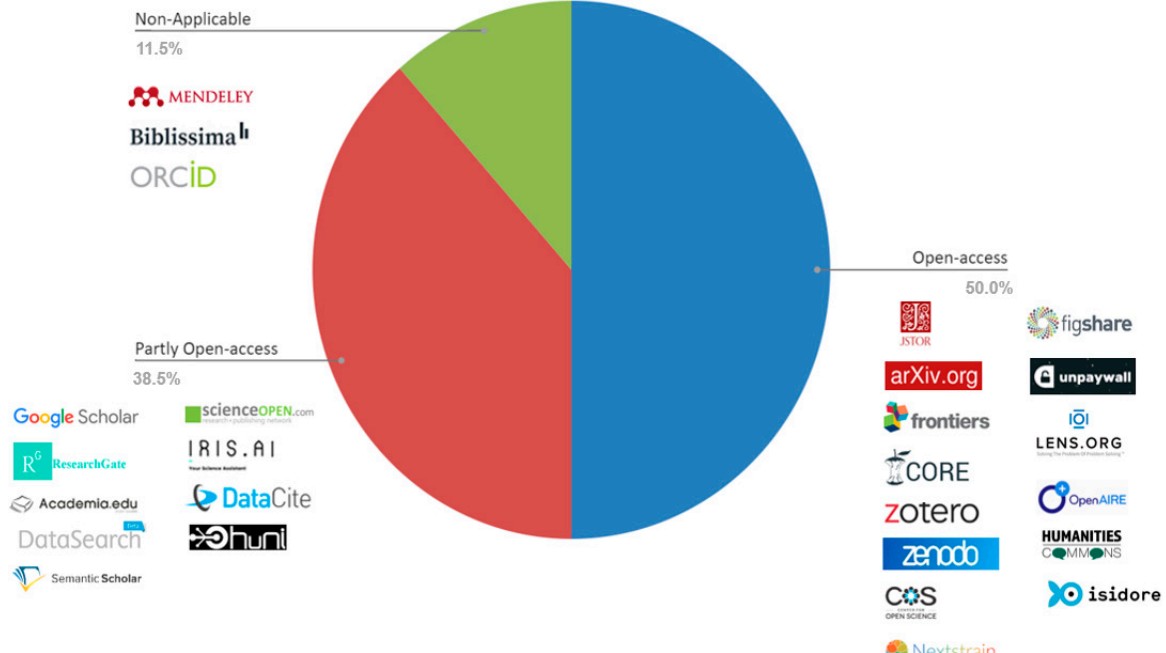

**Figure 1.** Analyzed platforms according to their status as "open access", "partly open-access" or "non-applicable".

*3.2. Method for Result 2: Interoperability of the GOTRIPLE Platform with the EOSC*

The EOSC is still under development and several stakeholders are involved in its building. This is quite a big issue for the TRIPLE consortium to develop a platform compliant with a system which is not complete and finished. Even if several requirements are now fixed, there are still a couple of issues and questions which require for the TRIPLE consortium to be as agile and flexible as possible. This explains why the technical requirements of GOTRIPLE, in the perspective of its interoperability, have been derived from monitoring and mapping of EOSC related projects and publications.

Initiated in 2016 by the European commission, the European Open Science Cloud (EOSC) has attracted great attention across Europe and worldwide. The ambition is to offer 1.7 million researchers and 70 million professionals in science, technology, the humanities and social sciences, a virtual environment with open and seamless services for storage, management, analysis and re-use of research data. It is TRIPLE's strategy to align the GOTRIPLE discovery platform design with the EOSC, so as to be visible through the EOSC platform and reach out to social sciences and humanities (SSH) researchers in Europe and beyond. In order to develop such a platform, the TRIPLE team adopted two methods which enabled us to have a comprehensive and up-dated view of the EOSC definition process and relevant results.

The release of EOSC-related outputs, which mainly follows the established roadmaps, is the result of a participatory process to which the TRIPLE's consortium—and especially the team of work package 6 ("Open Science and EOSC Integration")—are frequently asked to provide comments and feedback.

To be able to have a clear knowledge and understanding of the production of the main documentation relevant to TRIPLE's implementation as an EOSC service, a monitoring methodology has been established, by assigning a responsible partner to each of the EOSC working groups (WG) to report to the whole team about the specific advancements of the WGs.

The TRIPLE team also identified past and on-going EOSC-related projects that are relevant for the development of GOTRIPLE, and performed a monitoring and mapping exercise to have a complete vision of relevant deliverables, to be taken into account by TRIPLE's design, definition and implementation, documented in deliverable D6.1 "Report on the General Interoperability Requirements". For each of these projects, a list of relevant deliverables were identified. Each deliverable was then evaluated according to its relevance to the EOSC WGs, its overall purpose and main standards mentioned. The release date of the deliverable was also taken into account as some statements in deliverables may no longer be valid due to a natural evolution of the EOSC landscape over time.

Both activities have been considered a very useful exercise to analyze not only the outputs and deliverables individually, but also to evaluate the results in an aggregated manner, and to have a simultaneous overview of the results. The TRIPLE team will continue monitoring the appearance of further deliverables from the identified set of projects mentioned above as well as of newly funded projects, as this helps to develop to contextualize the GOTRIPLE technology, and to ensure the compliance with the common standards.

*3.3. Methods for Result 3: A Multilingual, Multidisciplinary and Multicultural Platform*

To take up the challenge of covering 27 disciplines and nine languages imply an overarching work related to data especially in a co-building process, i.e., by relying on the existent and the skills of the different partners. For this reason, the first phase of the project integrated tasks related to data retrieval and normalization to ensure a proper alignment of vocabularies whatever the language selected by the user. The broad scope of the disciplines (27 MORESS categories, Mapping of Research in European Social Sciences) covered by the platform required a strict methodology described below.

3.3.1. Advance Approach for Metadata Enrichment

The GOTRIPLE platform needs to handle various kinds of data from different resources and repositories. To have a clear overview of the types of data described and standards used in the platform,

methods have been developed for data access and exploitation, guidelines for intellectual property rights, ethics and privacy and disclosure risk management as well as data curation and preservation.

All the data aggregated in the GOTRIPLE platform follows a process of standardization, classification and indexing. The data must be organized in such a way as to meet the needs of researchers. In order to make them accessible via a search engine, the core of the platform is trained to identify keywords, titles and descriptions in each of the 27 identified disciplines (MORESS categories, developed by the Mapping of Research in European Social Sciences and Humanities project, https://cordis.europa.eu/project/id/HPSE-CT-2002-60060/fr) and in the nine languages. Around a hundred documents are therefore stored per discipline and language containing at least the following three metadata: Abstract, title and keywords.

The collected metadata are enriched using controlled vocabularies to improve their quality and their discoverability by using training machine learning algorithms based on scholarly publications (journals, books, articles) and metadata. This first process will then lead to the definition of a TRIPLE data model with links and description of the different relations between metadata. As shown in Figure 2, the enrichment consists of three different actions:

- classification based on a training scholarly article database and using advanced methods based on statistics and language analysis;
- normalization using thesauri;
- semantic annotations with a disambiguation tool using thesauri and the Wikidata database.

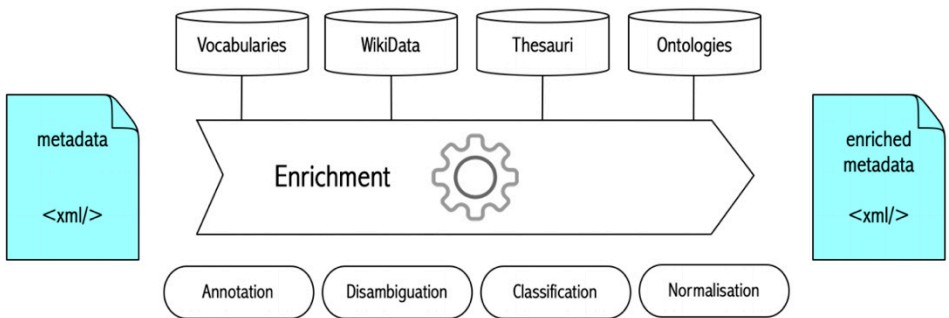

**Figure 2.** Metadata enrichment process.

### 3.3.2. Vocabulary Alignment

The first layer of the TRIPLE vocabulary comes from the Library of Congress Subject Headings (LCSH, https://id.loc.gov/authorities/subjects.html), which catalogs material held at the Library of Congress. It comprises a thesaurus, i.e., a controlled vocabulary of subject headings covering the social sciences and humanities (SSH) for use in bibliographic records. The methodology used for selecting the SSH-related concepts was based on identifying 14 basic concepts from the Frascati taxonomy (https://www.oecd.org/sti/inno/frascatiannexes.html) under SSH, then mapping these to 37 broad terms from LCSH and then extracting these and their children using the Linked Data API of the Library of Congress. Existing links of LCSH to other vocabularies were also imported from which labels in our nine target languages were extracted and added as labels in GOTRIPLE vocabulary's concepts. Moreover, existing LCSH links to wiki data have also been followed, from which more labels in our nine languages have been extracted and then added. The vocabulary is enriched as things progress with new concepts. Moreover, existing mappings from language-specific vocabularies and thesauri, such as the National Library of Florence (French) and Rameau thesaurus (Italian), have been processed in order to further enrich the multilingualism of the Triple vocabulary automatically. Missing labels are completed manually. A TRIPLE-specific guide describes the procedure for enriching the TRIPLE vocabulary (see Figure 3) with missing concepts as well as enriching the concepts with more labels in different languages, either manually or by leveraging existing mappings to LCSH.

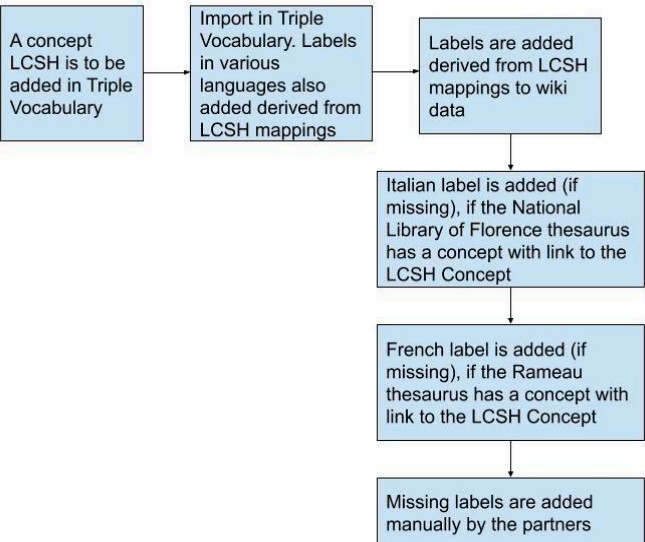

**Figure 3.** Schema of vocabulary enrichment.

### 3.3.3. Thesauri Alignment

One of the overarching issues for TRIPLE is related to the variety of thesauri in the SSH field as well as the diversity of European languages. Perfect alignment of thesauri within the same discipline across two different languages is difficult to achieve. This issue has a negative influence on potential collaborations between researchers and on the development of interdisciplinarity projects because the findability of data is heavily dependent on the quality of metadata and the alignment between thesauri. A specific methodology has been developed to cope with this challenge. It will support a process which can be reused for other fields and languages. Data providers need to be trained in the whole process of metadata enrichment in a multilingual and multidisciplinary context. This requires the following two steps: alignment of thesauri in each of the SSH disciplines and for each language, and process of enrichment of metadata through training and best practices report.

## 4. Materials

This last part of the paper presents the different materials that have been created to obtain the three preliminary results described at the beginning. To facilitate the understanding of the process, different materials are described together when they serve the same method and/or the same result.

### 4.1. Diversity of Materials for the Identification of GOTRIPLE Features

Identifying the main features of the GOTRIPLE platform is not an easy task. It depends on the partners but more so on the lessons learnt from the successes and failures of other platforms and on a good knowledge of the needs and constraints of future users. This is why two complementary methods have been used to achieve this goal with a diversity of materials.

### 4.1.1. Personas and Scenarios

Personas are "user archetypes" which can be used by designers to focus the process of design centering on the user. According to [12], personas "are not actual people but are synthesized directly from observations of real people". Personas are models and "precipitates" of real users obtained from user research, normally in the form of qualitative interviews or ethnographic observations. In other words, the personas are built out of qualitative data and encompass patterns emerging from across multiple interviews with end-users or ethnographies. A range of personas ($n = 8$) and scenarios ($n = 8$) have been produced from the analysis of the qualitative interviews to convey the user requirements to the technical partners, helping them to make design decisions. They also allow us to

more easily discuss what the platform functionalities will be with stakeholders, and they are useful during co-design workshops. Since co-design will enable the stakeholders to have an input into the design and functionality of the platform, the process also increases ownership and engagement with the final product. An example of one of the non-academic Personas is shown in Figure 1. It highlights how the platform could facilitate interactions between academic research and industry and other SMEs.

Shown in Figures 4 and 5 are examples of these personas. Mr David Green Figure 4 represents a non-academic stakeholder (a CEO of a small business), and Ms Carolina Weber (Figure 5) represents an academic stakeholder (a Ph.D. student).

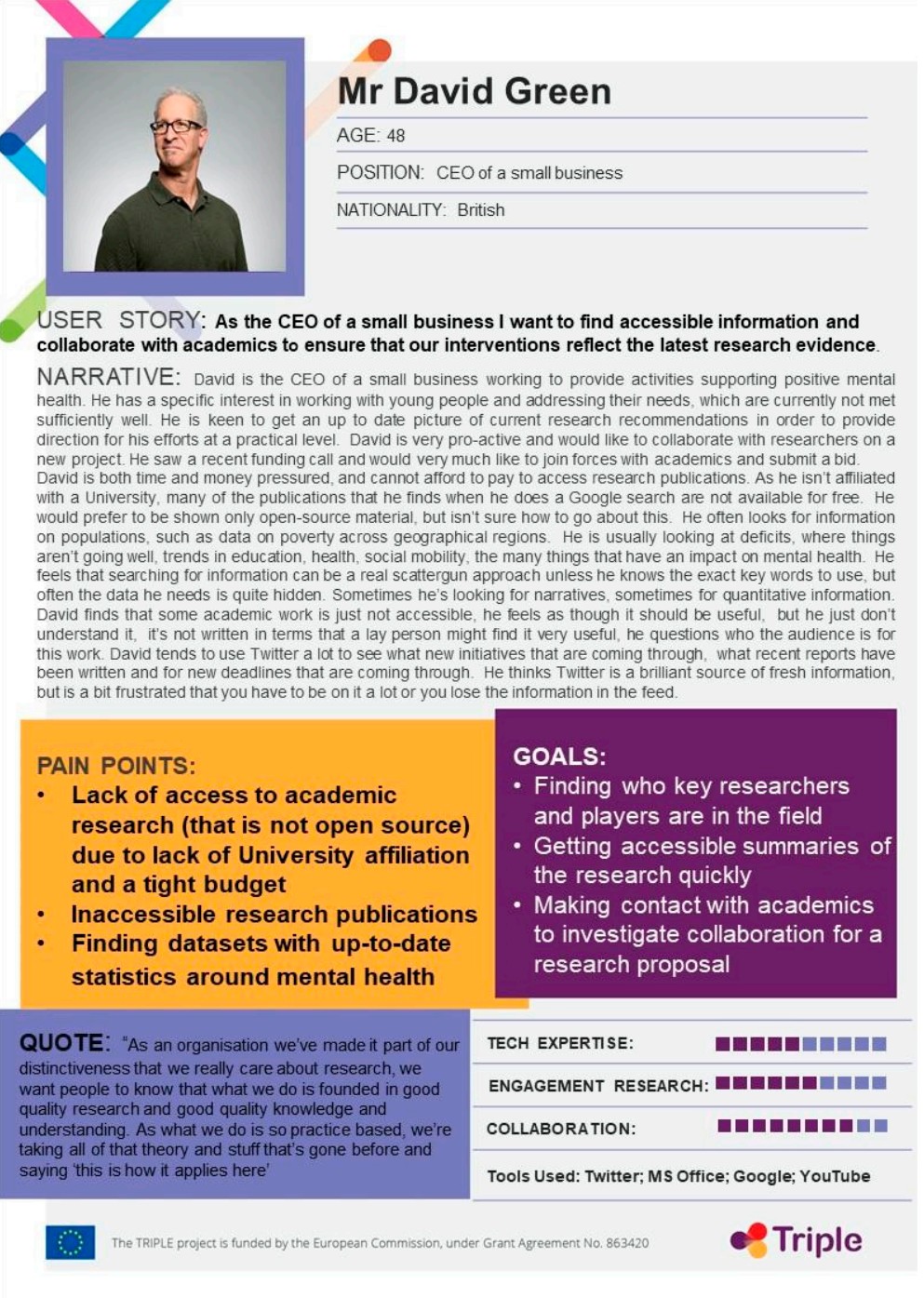

**Figure 4.** Example of a non-academic persona created from the results of the qualitative interviews and subsequent thematic analysis.

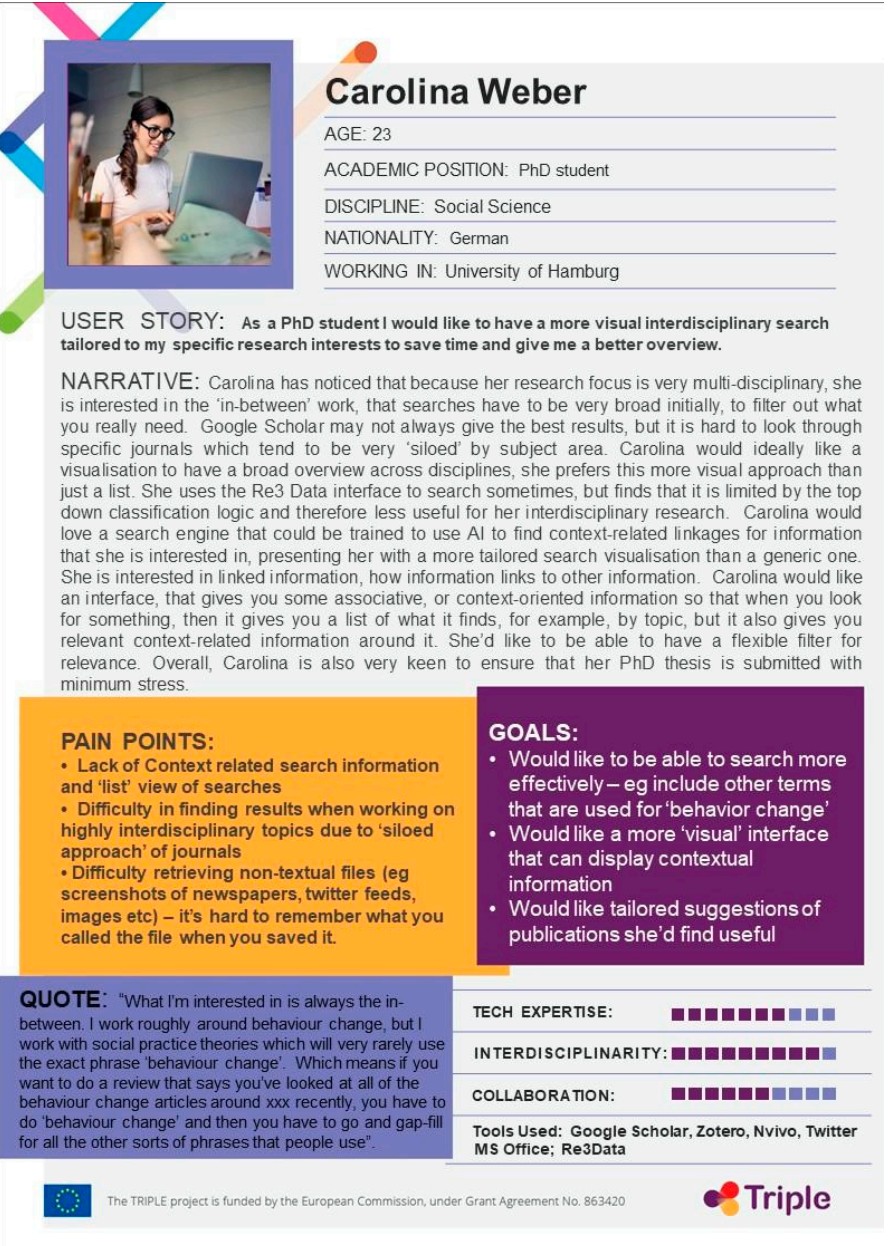

**Figure 5.** Example of an academic persona created from the results of the qualitative interviews and subsequent thematic analysis.

Scenarios can be simply seen as stories of the personas in the process of using the future product which is being designed (i.e., for example the narrative of a sociology researcher using the TRIPLE platform). Scenarios therefore are early prototyping tools which can support the designers in understanding better the user perspective toward using what is being designed and support the process of taking relevant decisions [12].

The main added value of scenarios is that from these, it is possible to derive high-level user needs or requirements. These should not be confused with requirements in software engineering, as the latter tend to focus on software functions more than on what the user does with e.g., a piece of software. User needs obtained from scenarios are generally the output of transforming the narrative scenarios into a series of steps that the persona does to achieve his/her goal within the scenario. In other words, the task is to translate the scenario into the precise list of actions that the persona does within the scenario narrative itself. In this way it is possible for designers to obtain a formal definition of the user

needs in the form of a list which can constitute the basis for the identification of functionalities and subsequent production of interface prototypes.

The non-academic scenario in Figure 6 highlights specific functions such as: Finding the key researchers in a specific area; Finding funding calls; Searching for projects; Searching for Academics and viewing their profiles; Viewing details of crowd-funding.

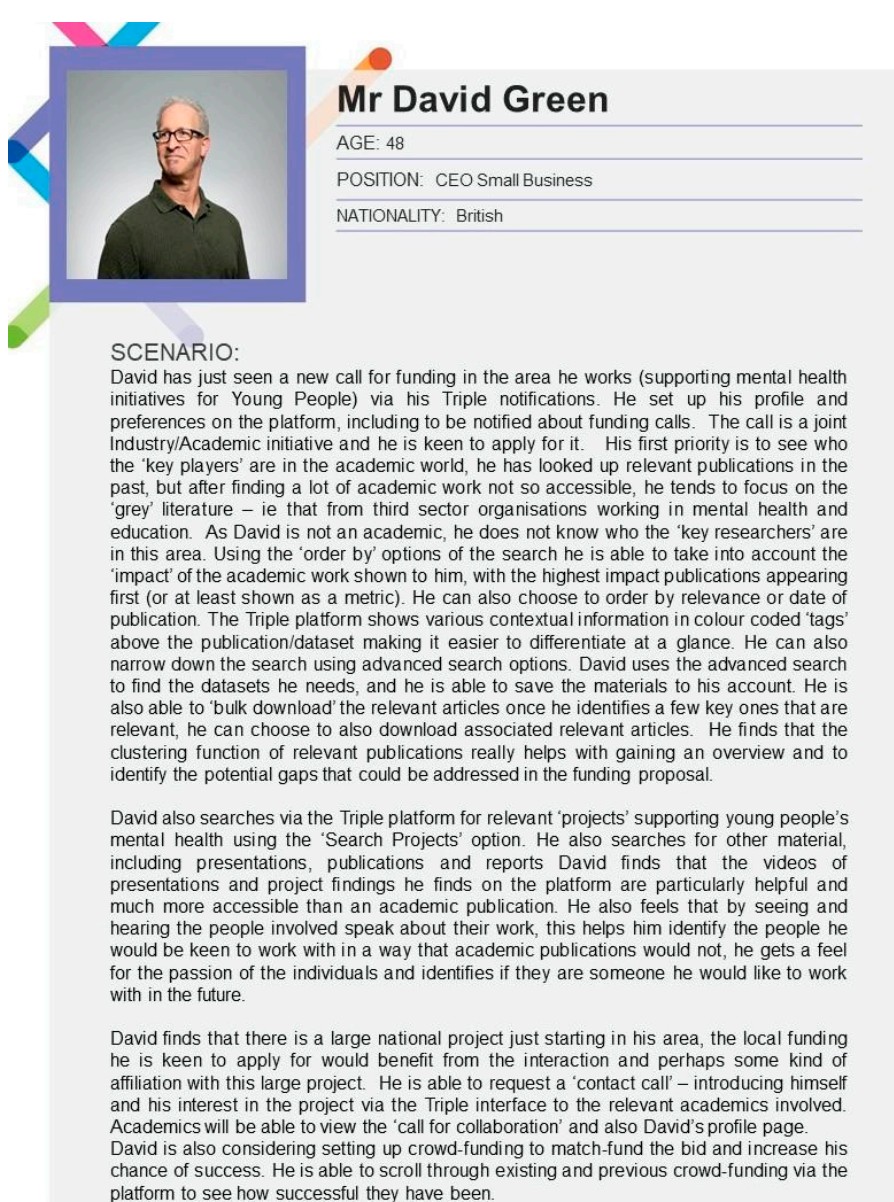

**Figure 6.** Example of a non-academic scenario created from the results of the requirement analysis.

The Scenario Steps obtained from David Green are:

7.1 The user shall be able to Search ordering by 'impact'

7.2 The user shall be able to Search by most recent publication

7.3 The user shall be able to Search for Projects

7.4 The user shall be able to Search for presentations (slides/video format)

7.5 The user shall be able to View academic profile

7.6 The user shall be able to see contact details of an academic

7.7 The user shall be able to View amount of funding crowd-funding calls obtained

The academic scenario in Figure 7 resulted in the following scenario steps

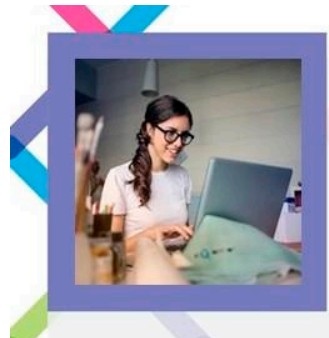

**Carolina Weber**

AGE: 23

ACADEMIC POSITION: PhD student

DISCIPLINE: Social Science/Politics

NATIONALITY: German

WORKING IN: University of Hamburg

SCENARIO:
Carolina is in her final year of her PhD, and is writing up her thesis. For the last few months she has been using the Triple platform to find information as it gives her more flexibility than other search platforms. She has enjoyed the artificial intelligence utilized to provide her with more tailored search results. She has provided feedback on the search results, and, over time these have become even better and now include results for research papers that do not include the specific keyword, but are related to them. This has meant a lot less time putting in different terms (as the different disciplines tend to use different terminology for very similar research. She finds that the 'silo effect' is much less now and she is shown papers from different disciplines rather than individual ones. As her research is at the intersection of different disciplines she found that many different terms were used by different researchers even though they were discussing very similar research.
Carolina has made good use of the article 'overview' feature, where a 5 point summary is made, highlighting the main points of the research paper without having to download it or read it all. She had previously seen this as a feature in Academia.edu, but it was a premium feature. Carolina often chooses to view her resources in the 'cluster' view where they are grouped with relevant articles together. She can also see which are the most important articles (icons displaying contextual information can be seen or hidden using the settings option). She finds the 'influential citations' or 'citation velocity' (as found in Semantic Scholar) are often more useful metrics than the overall number of citations. She often saves this 'cluster overview' and makes annotations and notes to help her gain an understanding of the 'state of the art' of the current research, enabling her to get a visual representation and a n overview of the research topic.
Carolina has been able to share her Twitter feed dataset with her PhD supervisor so that they could discuss how to process it together, she also made a visual plan for each chapter of her PhD, uploading this to her Triple private space. She has the option to share individual files or folders easily from the space without having to separately email the links. Having the option to 'tag' datasets and other non-textual files has made retrieval much easier and she also uses colour codes to ensure it's very easy to quickly identify documents relating to the different themes she is studying. She had previously used exclamation marks as a prefix to the file name to highlight the most important files in her collection, but Triple allows this 'importance rating' be done without having to rename the file. She finds it much easier to find the files she needs having all the visual cues that Triple allows, she downloads and saves a new article, choosing to add a 'tag' of 'digital behavior prompts' and the colour 'yellow' to the file (which she reserves for technology related articles) she also chooses to add the 'star' option to highlight it as important.
Open Access datasets are important for Carolina's work, she finds that Triple provides a much better way to find and store the datasets than her old way which took much longer, with Triple she can select the datasets to download and choose the best format to save it to her own space. Carolina feels that she is much more organized now, and that completing her thesis will be easier thanks to the Triple platform.

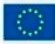 The TRIPLE project is funded by the European Commission, under Grant Agreement No. 863420  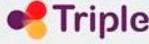

**Figure 7.** Example of an academic scenario created from the results of the requirements analysis.

Needs for Scenario 5: Carolina Weber

5.1 The user shall be able to Obtain tailored search results
5.2 The user shall be able to View an 'Article Overview' for a publication
5.3 The user shall be able to Share an individual file
5.4 The user shall be able to Share a folder
5.5 The user shall be able to Tag a dataset

5.6 The user shall be able to Color-code a file/dataset

5.7 The user shall be able to Download a single publication

5.8 The user shall be able to get an overview of a research topic

5.9 The user shall be able to get a visual representation of research topics

All of these functions, as well as further features raised thanks to other scenarios, then become the basis for a discussion about the necessary user needs and how to prioritize them for the building of the GOTRIPLE platform.

The identification of the priority of needs will also underpin the work on the design of the GOTRIPLE user interface. Moreover, the personas and scenarios produced can be used for other purposes, too. For instance, they can be integrated into communication material or be used during the co-design process in conjunction with the interface prototypes.

### 4.1.2. End-User Questionnaire

Following the work conducted for the definition of personas and scenarios for TRIPLE and based on qualitative interviewing, the questionnaire was planned with the intent of obtaining a much broader overview of the needs of the potential end users of the platform and to gain further knowledge to be used for the design. Among other sub-goals for the questionnaire there was the intent to measure, in more detail, the perception of end-users around discovery practices, networking practices, research tools and use and management of resources. Moreover, a final section of the questionnaire was prepared in a way to gain indication from the end-users about some of the directions that the TRIPLE platform could take to better meet the end-user needs. The questionnaire has gathered 925 responses from SSH researchers across Europe. The questionnaire data are being analyzed at the time of writing this paper and we still do not have clear results on the user needs that we can report here. We will report here thus on some of the demographics of respondents mainly.

The questionnaire has attracted responses from SSH researchers working across 26 European countries and other associated countries (such as Switzerland). The following Figure 8 shows the breakdown of responses per country, with some dominant countries such as France (*n* = 229), Portugal (*n* = 182), Italy (*n* = 101) and Germany (95) amounting to 65% of responses.

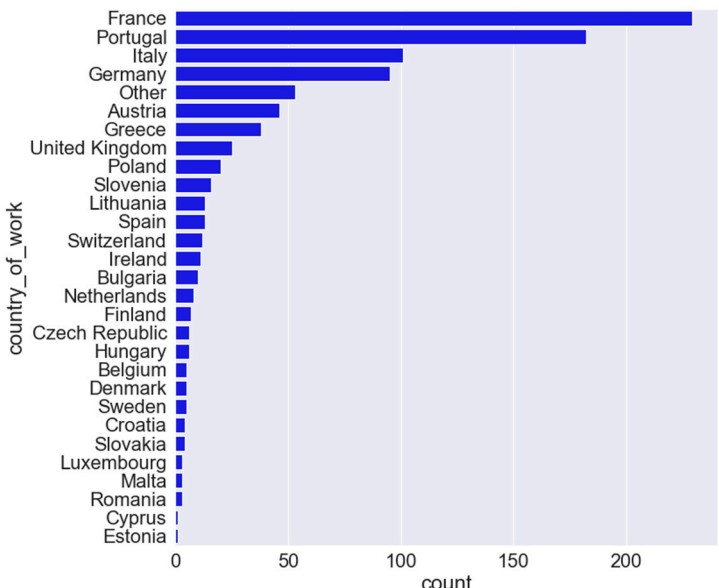

**Figure 8.** Country of work.

The questionnaire asked the respondents about their main methodological research approach/techniques, with the majority of respondents stating that they work mostly on qualitative

research (*n* = 475, 51%), followed by quali-quantitative (mixed-methods) researchers (*n* = 252, 27%) and quantitative (*n* = 177, 19%), with a minority selecting the other option and working with tangential techniques (e.g., Geographical Information Systems) (Figure 9). A note is that this distribution is not necessarily representative of the SSH community as a whole and it may be associated with the distribution channels of the questionnaire (for which we used several research mailing lists).

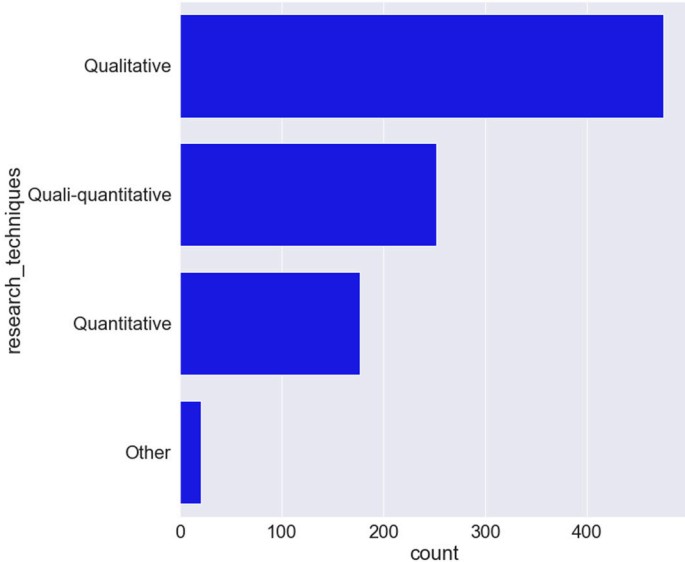

**Figure 9.** Main research techniques.

A further demographic we collected through our questionnaire relates with the main discipline of work of respondents. Figure 10 shows the responses with some dominant disciplines: Linguistics (*n* = 95, 10%), Sociology (*n* = 94, 10%), History (*n* = 74, 8%) and Library and Information Sciences (*n* = 65, 7%). Again, these results should not be seen as a reflection of the composition of the whole SSH research community and may be associated with the channels used for distribution of the questionnaire.

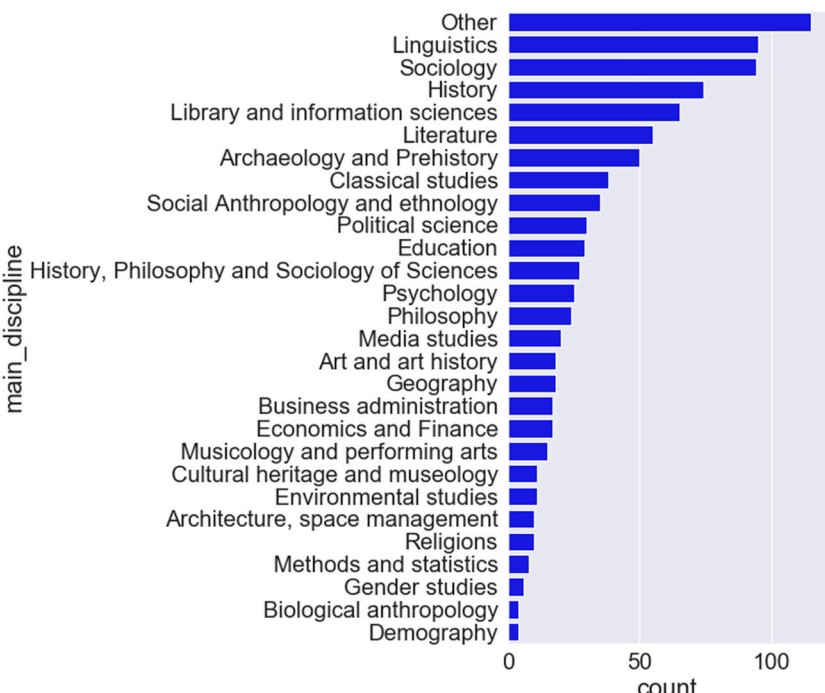

**Figure 10.** Main discipline.

One of the questions asked respondents to specify their priorities in terms of discovery that the TRIPLE platform should cover. There is a clear indication from respondents that their main discovery need is in the area of publications ($n$ = 626, 71.5%), followed by data ($n$ = 141, 16%). This gives a clear indication of the direction toward which the design should concentrate (Figure 11).

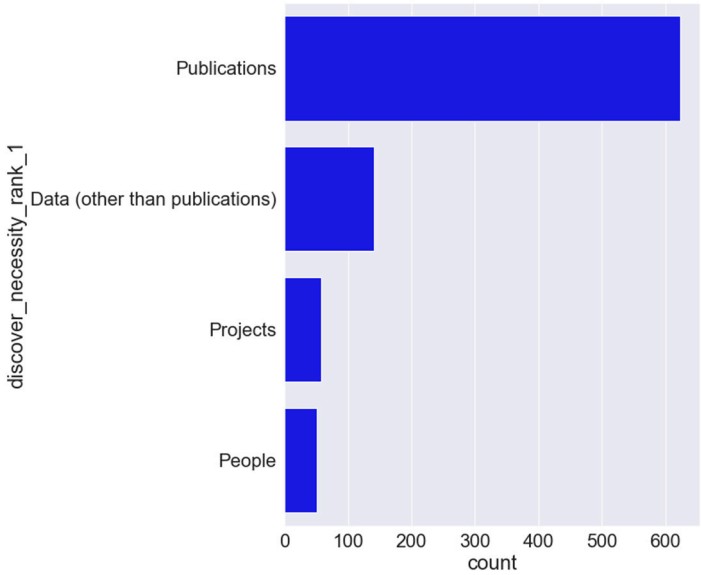

**Figure 11.** Main discovery need.

### 4.1.3. Market Analysis and Benchmark Activities

For the analysis of the 26 platforms, an analysis template was created that allows a structured documentation and a simple evaluation. The template (see Table 2) is structured according to a basic business model view. It describes what value the platforms offer their customers (value proposition), how this value is created (value creation) and how the platforms generate profits/cover costs from its activities (value capturing). Furthermore, the analysis also covers organizational insights (board, team, legal form, etc.), strengths and weaknesses as well as impressions on usability and user experience. The twenty-six completed analysis documentations were transferred into a summary table and evaluated through qualitative content analysis.

To analyze whether competitors offer a similar feature set as TRIPLE (see Table 3), a table with the following dimensions was created: Search and filter systems, recommendation system, social services, annotation tools, and visual discovery. The search and filter system dimension refers to the ability to search for research outputs, and use filters to further define the search criteria. The recommendation dimension indicates if the platform offers recommendations of e.g., research outputs, projects, and authors. The social dimension refers to features that leverage social interaction within the platform e.g., share articles, create groups, collaboration work. The annotation dimension refers to the availability of tools that allow users to annotate research outputs within the platform. Lastly, the visual discovery dimension refers to visual search and discovery interfaces.

Looking at the functions and services provided by the competitors, we recognize that the planned feature-set for the GOTRIPLE platform (such as the conjunction of the visualization tool, annotation tool, trust building system, recommender system and crowdfunding service) represents unique features that will distinguish TRIPLE from the competition. In this perspective, this service provider approach has reinforced the conclusions of the co-design process.

**Table 2.** Competitor Analysis Template.

| **BASIC DESCRIPTION** |
| --- |
| Platform Name:　　　　　　Site URL:　　　　　　Platform Logo: |
| Origin of platform provider/operator. |
| Short description of platform (Mission, Vision, etc.) |
| Overview on Offerings (Services, Products, Features, Functions) |
| Size of the platform (Number of users/documents etc.) |
| Focus (regional, geographic, specific research domains or target groups, language versions) |
| Organization Insights (board, team, legal form etc.) |
| Financing (type of revenue streams, budget figures, cost factors) |
| Marketing/Dissemination |
| Partners and Stakeholders |
| **DETAILED CONTENT DESCRIPTION** |
| Most relevant functions and features (Please indicate main functions and features and describe with screenshots and short explanations) |
| Value add of platform for stakeholders (What feature/function is unique/outstanding? What add on benefits does the platform offer? How would you describe the Unique Selling Proposition of the platform?) |
| **USABILITY/UX** |
| Clearly and understandable symbols and wording? |
| User Orientation—Can I navigate within the platform with relative ease? |
| Design of user interface—Clear arranged, not confusing. Is the interface well organized, logically laid out, ease to navigate—or is it the opposite (cluttered, illogical, complicated)? |
| User motivation—Are users motivated to use the platform more often, if yes, how? |
| Learnings concerning usability/UX for TRIPLE—What should we transfer to TRIPLE, what should we avoid? |
| **SUMMARY and CONCLUSION** |
| Strengths and Weaknesses |
| Personal rating (1 = very bad, 10 = best in class) |
| Relevance for consideration within TRIPLE—What can we learn or should take into consideration for TRIPLE Platform? |

**Table 3.** TRIPLE's main features compared to competition.

| Type | Competitor Platform | Search and Filter | Recommendation | Social | Annotation Tools | Visual Discovery |
| --- | --- | --- | --- | --- | --- | --- |
| Academic search engine | Google Scholar | Yes | Yes | Partly | | |
| | Elsevier Data Search | Yes | | | | |
| | Semantic Scholar | Yes | Yes | | | |
| | Frontiers | Yes | Yes | | | Partly |
| | Lens.org | Yes | Yes | | | |
| | Iris.ai | Yes | | | | Yes |
| | Isidore | Yes | Yes | Partly | | |
| Search engines and directories for OA resources | JSTOR | Yes | Yes | | | |
| | arXiv.org | Yes | Yes | Partly | Partly | |
| | CORE | Yes | Yes | | | |
| | unpaywall | Yes | | | | Yes |

**Table 3.** *Cont.*

| Type | Competitor Platform | Search and Filter | Recommendation | Social | Annotation Tools | Visual Discovery |
|---|---|---|---|---|---|---|
| Science-oriented social media | Researchgate | Yes | Yes | Yes | | |
| | Academia.edu | Yes | Yes | Yes | | |
| | Mendeley | Yes | Yes | Yes | Yes | |
| | Zotero | Yes | | Yes | Yes | |
| | Humanities commons | Yes | | Yes | Yes | |
| Dissemination platform | Center of Open Science | Yes | | Yes | | |
| | ScienceOpen | Yes | | Yes | | |
| Repositories - institutional or subject | zenodo | Yes | | Yes | | |
| | figshare | Yes | | Partly | | |
| | OpenAIRE Explore | Yes | | | | |
| Multidisciplinary academic databases | ORCID | Yes | | | | |
| | DataCite | Yes | | Partly | | |
| | Nextstrain | Yes | | | | Partly |
| Library catalogues and discovery systems | huni | Yes | | | | Partly |
| Shadow library | Biblissima | Yes | | | | Yes |

*4.2. Materials for the Interoperability of the GOTRIPLE Platform with the EOSC*

EOSC WGs outputs and relevant official documents are the materials WP6 (Work Package dedicated to the integration into the EOSC) worked with in order to have a clear vision of the EOSC definition and to design TRIPLE integration into it.

The general reference document is the EOSC General Interoperability Framework, released as a draft version open for comments in May 2020. The other main materials come from the FAIR and the Architecture working groups, and are presented in Table 4. They list 8 reports, focused on the main architectural and technical requirements to be followed. Each of these reports have been analyzed and commented in order to keep the main important points for GOTRIPLE development.

**Table 4.** Main relevant outputs from the Findable, Accessible, Interoperable and Re-usable (FAIR) and the Architecture European Open Science Cloud (EOSC) Working groups.

| EOSC Working Group | Analysed Reports |
|---|---|
| FAIR | Turning FAIR into reality |
| | The final report and action plan from the European Commission Expert Group on FAIR Data of 2018, setting up the conditions to data FAIRness. |
| | FAIR metrics for EOSC (Provisional) (February 2020) |
| | The document reports on the activities of the RDA (Research Data Alliance) WG (Working Group) on the FAIR data maturity model, the FAIRsFAIR project, and more focused works (e.g., FAIR software). The FAIR metrics and the FAIR assessment tools are intended to guide progression towards FAIRness—which partly contradicts the fact that the FAIR metrics will also be part of the FAIR certification: are the FAIR metrics an auto-assessment tool or a technical requirement to be part of the EOSC? The report contains a list of FAIR data indicators which will be detailed by the WG in a future work. |
| | EOSC service certification for FAIR outputs (Provisional) (February 2020) |

**Table 4.** *Cont.*

| EOSC Working Group | Analysed Reports |
|---|---|
| | The draft report mainly suggests using the CoreTrustSeal certification for repositories as a tool to build a FAIR ecosystem. The certification could then be used to establish a «European Network of trustworthy repositories». It is planned to combine the certification with the FAIR metrics. The document contains reports on various workshops and surveys organized by the FAIR WG and the project FAIRsFAIR which all seem to have a very provisional nature. |
| | PID (Persistent Identifier) (policy for EOSC (Second version) (May 2020) |
| | The draft report (final version planned for October 2020) provides definitions and recommendations for a sustainable PID infrastructure. It contains details on technical requirements, distribution of roles, and governance. The link with FAIR principles, and more precisely FDOs (FAIR Digital Object), is explicit. Not very precise is the nature of the "PID infrastructure" itself, partly because the actual target audience of the policy is unclear, partly because the responsibility of EOSC as a legal entity in this context is mentioned but not defined. |
| | Recommendations for Services in a FAIR Data Ecosystem (August 2020) |
| | The document reports on workshops co-organized by FAIRsFAIR, RDA Europe, OpenAIRE (European Open Science Infrastructure, for open scholarly and scientific communication), EOSC-hub, and FREYA. The recommendations address the FAIR principles from an ecosystemic point of view, considering that there is a lot of activity around the concept of FAIR data "but it is much less clear what should be expected from a data service in the FAIR data ecosystem". The report thus analyzes the gaps, both for each actor of the ecosystem and between each of them (researchers, data stewards, service providers, etc.). A first workshop was held for "service providers and research infrastructures", a second one with "research support staff and researchers", each group defining its own recommendations. A third workshop established a prioritization process of the recommendations. Interestingly, the report notes that different skills have to be combined to realize a FAIR ecosystem (technical/domain expertise), and that there are some discrepancies between the "Turning FAIR into reality" report and the communities priorities, thus suggesting that the official roadmap for FAIRification could be reshaped through their insights. |
| | EOSC AAI (Authentication and Authorization Infrastructure) First Principles (April 2020) |
| | This report identifies three principles for EOSC AAI: (1) User experience is the only touchstone; (2) All trust flows from communities; (3) There is no center in a distributed system. |
| | These principles clearly state that the design of the EOSC AAI will be user centered, and the implementation will be a distributed architecture. |
| | EOSC AAI Architecture 2019 (June 2020)—This is a draft report, currently shared internally among the Working Group Members. |
| | This report captures the current status of the EOSC AAI architecture discussions that are based on the AARC Blueprint Architecture 2019 (Authentication and Authorisation for Research and Collaborations). It also identifies the challenges and the areas that require further work. |
| | The potential benefits are: Being a GOTRIPLE user, s/he can also access EOSC services. On the other hand, EOSC SSH (Social Sciences and Humanities) researchers and other Science communities' users by default become GOTRIPLE users and are able to use the GOTRIPLE platform—this will enlarge the TRIPLE user-base and make TRIPLE more visible to European science communities. |
| | PID Architecture (draft) (June 2020) - |
| | *This is a draft report, currently shared internally among the Working Group Members.* |
| | This report describes the main components of a global PIC architecture microcontrollers of memory organization (ram,rom,stack), and the PID registration and resolution framework. It discusses some existing technology for implementing such a PID framework, and examples of the PID services. |
| | In GOTRIPLE, ORCID identifier (Open Researcher and Contributor ID) is adopted for data registration and processing, which is interoperable with the proposed EOSC PID Architecture. TRIPLE also closely interacts with relevant EOSC projects such as FREYA, a 3-year project collaborating with OpenAIRE Advance and EOSC-hub and focusing on developing a PID infrastructure for EOSC. |

WP6 partners have carried out a mapping exercise of relevant deliverables produced by the main EOSC related projects in order to provide a comprehensive analysis of the EOSC interoperability requirements. The analysis was aimed to understand the developments of the EOSC environment in terms of interoperability and at the same time to understand which public deliverables have to be taken into consideration for the overall project development in TRIPLE.

A total of 11 projects have been investigated and 38 relevant deliverables have been thoroughly analyzed. The full results are available as an Annex of Deliverable 6.1—General Interoperability requirements, submitted to the European Commission at the end of September 2020.

Table 5 offers the highlights of 15 deliverables from four projects, particularly relevant for TRIPLE.

**Table 5.** Mapping EOSC-related projects and relevant deliverables.

| Project | Analysed Deliverables |
| --- | --- |
| EOSC-hub brings together multiple service providers to create the Hub: a single contact point for European researchers and innovators to discover, access, use and reuse a broad spectrum of resources for advanced data-driven research. | *Deliverables related to Architecture WG*<br>D4.2 Operational Infrastructure Roadmap—*relevant as it describes the guidelines for the actions that are to be taken in order to ensure interoperability at the level of EOSC-hub service catalogue which can be taken as lessons learned for the work in TRIPLE*<br>D5.3 1st Report on maintenance and integration of federation and collaboration services<br>D6.2 First report on the maintenance and integration of common services<br>D7.2 First report on Thematic Service architecture and software integration<br>D10.3 Technical Architecture and standards roadmap v1—*relevant as it gives examples how Research Enabling services benefit from diverse features of Access Enabling services when being incorporated within a unified Hub.*<br>D10.4 EOSC Hub Technical Architecture and standards roadmap v2—*relevant for the TRIPLE plans for managing researchers' identity*<br>D10.5 Requirements and gap analysis report v1 |
| FREYA is a 3-year project funded by the European Commission under the Horizon 2020 program. The project aims to extend the infrastructure for persistent identifiers (PIDs) as a core component of open research, in the EU and globally. FREYA will improve discovery, navigation, retrieval, and access to research resources. | D2.1 PID Resolution Services Best Practices—*relevant for WP2 (Work Package dedicated to data acquisition) and WP4*<br>D3.1 Survey of Current PID Services Landscape—*relevant for WP2, especially to discuss the needs of a TRIPLE ID*<br>D3.2 Requirements for Selected New PID Services—*relevant for TRIPLE WP2 and WP5, especially for the links to innovative service* |
| OpenAIRE-Advance continues the mission of OpenAIRE to support the Open Access/Open Data mandates in Europe. By sustaining the current successful infrastructure, comprising a human network and robust technical services, it consolidates its achievements while working to shift the momentum among its communities to Open Science, aiming to be a trusted e-Infrastructure within the realms of the European Open Science Cloud. | D 4.2—A multi-module Open science kit—*relevant for Task 6.3 as a preliminary work on Open Science training*<br>D 7.3. Interoperability with Research Infrastructures—relevant as it highlights how the work that focuses on services built on the basis of Open Science publishing practices supports cross-community communication and collaboration. Moreover, this deliverable allows the drawing of a distinction between the OpenAIRE services and the implemented and envisaged ones of the TRIPLE project. |
| EOSC Enhance project is committed to improve the EOSC Portal by making it the added value one-stop shop/entry point for the EOSC users and stakeholders, by enabling easy access to EOSC resources such as services, data, scientific products and other resources to European scientists. | D 2.2 EOSC Processes Development and Consensus<br>D 2.4: EOSC Service Catalogue Analysis—*relevant for TRIPLE because it facilitates the discoverability of EOSC resources across disciplines*<br>D 3.1: EOSC Portal Functional and Non-Functional Specifications<br>D 3.2: EOSC Portal Open APIs Specifications of Service and Resources Providers—*relevant for TRIPLE as it shows the requirements needed to be integrated in the EOSC portal* |

All these analyzed materials contribute to define TRIPLE's positioning into the EOSC context from an architectural point of view.

### 4.3. Materials for Having a Multilingual, Interdisciplinary and Multicultural Discovery Platform

Two main materials can be used to ensure the multicultural and interdisciplinary vocation of GOTRIPLE: the data acquisition plan and the data management plan. However, these materials can also be seen as the first concrete results of the TRIPLE project (i.e., deliverables). Indeed, the TRIPLE thesaurus has been elaborated from different materials such as the existing vocabularies or the existing SSH categories. One of the goals of the data acquisition plan was to identify the different practices of SSH repositories to select the most appropriate ones for GOTRIPLE.

The TRIPLE data acquisition plan contains the technical specifications to be implemented in order to collect metadata about the research outputs from Social Sciences and Humanities in the nine covered languages (Croat, English, French, German, Greek, Italian, Polish, Portuguese and Spanish). It defines the process of collecting metadata until their exposition in the TRIPLE database through a two-fold approach: (1) Metadata provision by processing chains of aggregation platforms and (2) Semantic enrichment and resource linking by the TRIPLE pipeline. A delivery platform will be the communication interface between both processes (Figure 12).

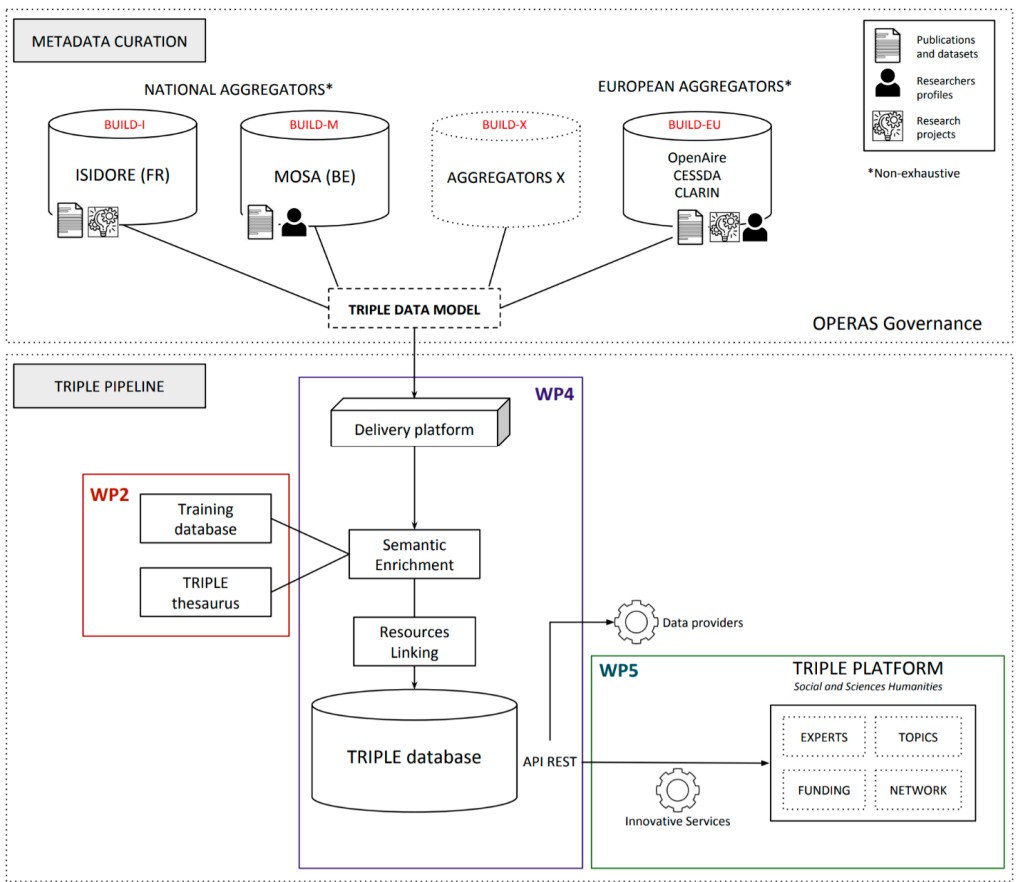

**Figure 12.** Schematic and simplified representation of the two-fold approach to ingest metadata in the TRIPLE database.

As a first phase, metadata are collected by aggregation platforms which are part or out of the consortium and dropped on the delivery platform. To collect and expose their metadata, these platforms use generic processing chains called BUILD. In accordance with the TRIPLE recommendations and with their agreement, the BUILD chains will deliver selected metadata on a delivery platform, under the monitoring of a scientific advisory committee. This implies that the project creates a model, called the TRIPLE data model, that the aggregation platforms might align with to be compliant with the discovery platform. To start the project, the ISIDORE platform, developed by the coordinator of the

project, had been chosen to be the first source of metadata, by using its processing chain "BUILD-I" (Figure 13). In the long run, to reach a satisfying level of exhaustivity, other BUILD chains will be added to cover the maximum of resources available in the whole SSH community worldwide. In a second phase, by a connection to the delivery platform, the TRIPLE pipeline will be able to collect, enrich and link the metadata corresponding to the three types of resources targeted by the project.

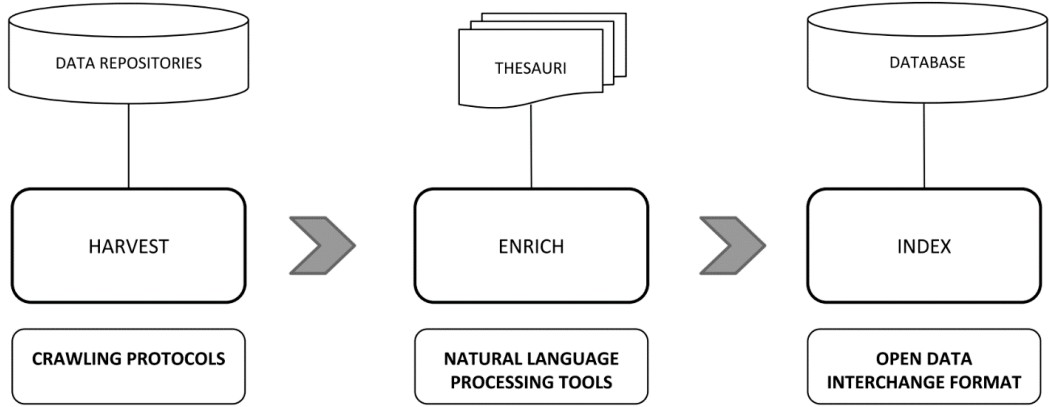

**Figure 13.** A generic processing chain.

The semantic enrichment will imply the creation of a TRIPLE thesaurus to align the vocabularies in the nine languages. The enriched and linked metadata will be then both stored in a tripleStore and indexed in the TRIPLE database and available through REST APIs (Representational State Transfer) for the Innovative Services to run their tools or for data providers to retrieve improved metadata.

The data acquisition plan has also detailed the TRIPLE data model (Figure 14) for each harvested resource (research data, projects and profiles) in order to first harmonize the kind of metadata needed for the discovery platform but more to plan the necessary linking between the different resources as presented in the following schema.

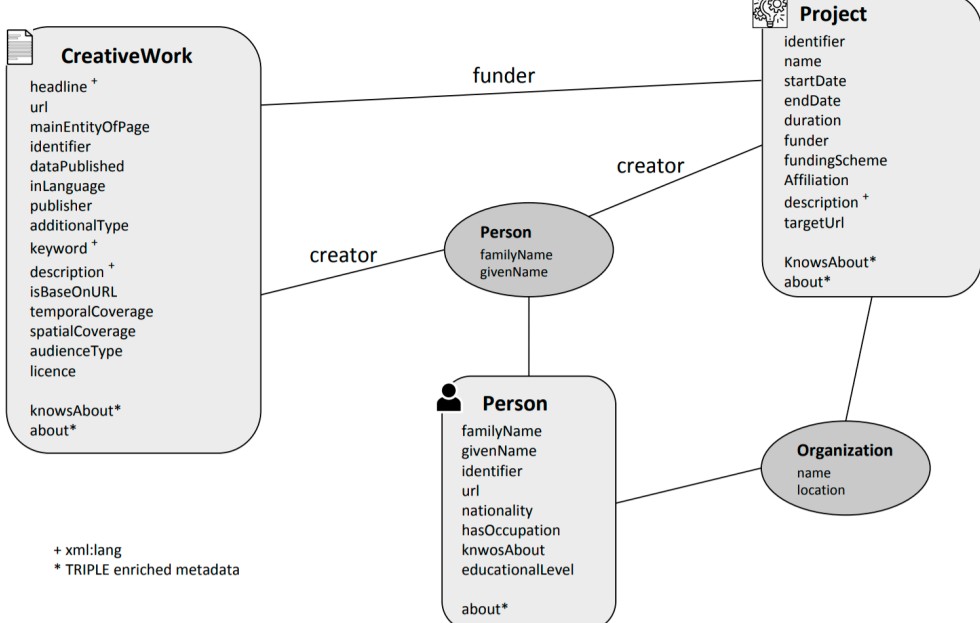

**Figure 14.** TRIPLE data model and linking between the 3 types of resources. Legend: "CreativeWork" for research documents publications and datasets, "Project" for research projects and "Persons" for researcher profiles.

The Data Acquisition Plan sets out an ambitious blueprint for aggregating Social Science and Humanities data descriptions on a vast scale, in order to make many disparate data collections searchable and thus easily accessible to researchers via a single portal, which will constitute a part of EOSC. It provides a detailed approach in two phases to collect metadata in order to achieve this ambition. It provides a strong material to contribute to the building of a multilingual and interdisciplinary platform.

## 5. Conclusions

The TRIPLE project gathers 19 European partners and has a duration of 42 months. It has to be both innovative and to rely on existing infrastructures and resources. For this kind of project to succeed is a challenge, and the progress needs to be measured step by step and to rely on different methods. This explains why the three main preliminary results presented in this paper have been achieved by such a diversity of methodological approaches. It is important to keep in mind that these are preliminary results, which can become, in turn, the basis for the further development of GOTRIPLE (iterative approach). In fact, the diversity of methods and materials reflects the willingness of the consortium to foster diversity of scientific practices and communities. This is one of the most important points to raise: the user-centric approach is not deployed only for the potential of the future GOTRIPLE users, but also of the consortium in itself. In this perspective, we believe that ICT technologies contribute to the coordination and federation of the diversity of SSH practices without diminishing their differences.

**Author Contributions:** Conceptualization, S.D., E.B., G.B., Y.C., F.d.D., P.F. and S.P.; Methodology, S.D., E.B., G.B., Y.C., F.d.D., P.F. and S.P.; Validation, S.D., E.B., G.B., Y.C., F.d.D., P.F. and S.P.; Formal Analysis, S.D., E.B., G.B., Y.C., F.d.D., P.F. and S.P.; Investigation, S.D., E.B., G.B., Y.C., F.d.D., P.F. and S.P.; Resources, M.E., H.G., A.G., E.G., Y.M., S.d.P., C.P. and E.T.-C.; Writing Original Draft Preparation, S.D., E.B., G.B., Y.C., F.d.D., P.F. and S.P.; Writing Review & Editing, S.D., E.B., G.B., Y.C., F.d.D., P.F. and S.P.; Visualization, S.D., E.B., G.B., Y.C., F.d.D., P.F. and S.P.; Supervision, E.B., F.d.D., G.B., S.P., S.D., Y.C. and P.F.; Project Administration, S.D. and E.B.; Funding Acquisition, S.D. All authors have read and agreed to the published version of the manuscript.

**Funding:** The project has received funding from the European Union's Horizon 2020 research and innovation programme under grant agreement No 863420.

**Conflicts of Interest:** Authors declare no conflict of interest.

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
