# Peer review of "GOTRIPLE: A User-Centric Process to Develop a Discovery Platform"

_information, doi:10.3390/info11120563_

Round 1

Reviewer 1 Report

Excellent results and a very useful overview of co-design process and co-building. 

Author Response

Thank you for your valuable comments.

Reviewer 2 Report

The title, although promising at first, does not match with the manuscript content, and may even be misleading. The authors should clarify where is the community which drives the platform.

The structure and format of document are more aligned with the characteristics of a project seeking funding, written by professional consultants. Its structure does not match with a standard research paper.  The document details the platform’s business model, explaining which features distinguish TRIPLE from the competition, describing potential future interoperability with the EOSC platform, FAIR principles fulfilment, and so on.  However, no empirical research is included. The definition of the main features of the platform, its potential interoperability and multilingual, multicultural and interdisciplinary vocation cannot be considered empirical results.

Taking the above into account, it seems like the paper is being used to publicize the TRIPLE platform.  Additionally, the document does not address the question of how much citizens, such as Mr. David Green or Carolina Weber, would have to pay to access the TRIPLE platform, or who should be responsible for paying for such access.

Author Response

Response to Reviewer 2 Comments

Point 1:  The title, although promising at first, does not match with the manuscript content, and may even be misleading. The authors should clarify where is the community which drives the platform.

Response 1: We agree with your comment, and we have renamed the document as follows : GOTRIPLE: a user-centric process to develop a discovery platform

Point 2 : The structure and format of document are more aligned with the characteristics of a project seeking funding, written by professional consultants. Its structure does not match with a standard research paper.  The document details the platform’s business model, explaining which features distinguish TRIPLE from the competition, describing potential future interoperability with the EOSC platform, FAIR principles fulfilment, and so on.  However, no empirical research is included. The definition of the main features of the platform, its potential interoperability and multilingual, multicultural and interdisciplinary vocation cannot be considered empirical results.

Response 2: We also fully agree with the remark that our paper is not a standard research one. The framework requested for this article is restrictive and does not respond to the practices of SSH articles. We had to adapt to a framework coming from the STEM which does not fit both with SSH and with the subject raised. The imposed sections of the format have been a problem from the outset because our project to create an SSH discovery platform with innovative services does not exactly meet the requirements for writing a scientific article that wants to highlight empirical results. In our opinion, this is because in the case of the humanities and social sciences, the scientific path is not identical to the path of the so-called hard sciences. As an asset we wanted to take up this challenge and face this difficulty in order to approach our project from a new angle. The project takes up different challenges such as for example creating a complete platform with several types of data and complex functionalities to help research in SSH which does not constitute empirical results for all that. Our aim in this paper is to highlight more the methodology used to build this platform in the context of the European Open Science Cloud.

Point 3 : Taking the above into account, it seems like the paper is being used to publicize the TRIPLE platform.  Additionally, the document does not address the question of how much citizens, such as Mr. David Green or Carolina Weber, would have to pay to access the TRIPLE platform, or who should be responsible for paying for such access.

Response 3: Concerning pricing of the platform, there is nothing set in stone. We are still in the ideation phase of our business model. For the time being we do not see a big potential at the beginning (launch of platform) to charge users for using the ordinary discovery functions. Since we do not have yet an overview on how users accept and demand the functions and features of GOTRIPLE it is hard to come up with prices and charging. But of course, we could imagine charging users for premium features but this is a bit too early to fix this. A working group is dedicated to the sustainability and business model of the model and is in charge of delivering tangible results over the next year.

Reviewer 3 Report

- General observations: this is a very complete and detailed report on a project, rather than a report on a specific investigation. The project includes various investigations, which are not presented here in full, but in the form of their main parameters and their main results. We therefore understand that this article is not a standard research report, but it is a valuable report that should be published, for the benefit of the public of this journal and of professionals in the digital humanities sector.   - Language registration: excessively promotional in the initial descriptive sections of the project, such as in the commercial documentation of a company's product. All the text should be carefully checked to avoid this registration. - This more commercial than academic record is very visible in the description of Result1: it presents a description with little reference to operational aspects, using a rather metaphorical record. The results should be presented in another more academic language registry based on detailed and specific functions. - Competitors: the absence of Microsoft Academic and H-Net is surprising. - Query interface. The absence of information on the inquiry form or on the functions and design of the inquiry interface stands out. If the product is aimed at researchers, it must incorporate an advanced search with complete search functions, including Boolean operators, search by fields, etc.

Author Response

Point 1:  General observations: this is a very complete and detailed report on a project, rather than a report on a specific investigation. The project includes various investigations, which are not presented here in full, but in the form of their main parameters and their main results. We therefore understand that this article is not a standard research report, but it is a valuable report that should be published, for the benefit of the public of this journal and of professionals in the digital humanities sector.   

Response 1:  First of all, we would like to thank the reviewers for their attentive relecture and remarks. The latter are important feedback for the consortium. Irrespective of comments on form, comments on substance help us to ask the right questions, to distance ourselves from the technical tasks we are currently preoccupied with and thus to put the objectives of the platform and the development strategy into perspective. We realise that we lacked clarity on such essential matters as the possible cost of the platform or the design of the search engine. We fully agree with the remark that our paper is not a standard research one. The framework requested for this article is restrictive and does not respond to the practices of SSH articles. We had to adapt to a framework coming from the STEM which does not fit both with SSH and with the subject raised. The imposed sections of the format have been a problem from the outset because our project to create an SSH discovery platform with innovative services does not exactly meet the requirements for writing a scientific article that wants to highlight empirical results. In our opinion, this is because in the case of the humanities and social sciences, the scientific path is not identical to the path of the so-called hard sciences. As an asset we wanted to take up this challenge and face this difficulty in order to approach our project from a new angle. The project takes up different challenges such as for example creating a complete platform with several types of data and complex functionalities to help research in SSH which does not constitute empirical results for all that. Our aim in this paper is to highlight more the methodology used to build this platform in the context of the European Open Science Cloud.

Point 2: Language registration: excessively promotional in the initial descriptive sections of the project, such as in the commercial documentation of a company's product. All the text should be carefully checked to avoid this registration. 

- This more commercial than academic record is very visible in the description of Result 1: it presents a description with little reference to operational aspects, using a rather metaphorical record. The results should be presented in another more academic language registry based on detailed and specific functions. 

Response 2: Now considering remarks related to the form as too promotional writing, we agree as it is damaging to the scientific character of the article. We have therefore taken care in the corrected version to reduce as much as possible the use of overly commercial phrases.

Point 3: Competitors: the absence of Microsoft Academic and H-Net is surprising. 

Response 3: The absence of Microsoft and H-Net as competitors is due to the use of the Alexa rank score (www.alexa.com) to determine the platforms popularity. This ranking system considers the estimated average of daily unique visitors and the number of page views over the past three months. The top ten ranked platforms were included in the competitors’ analysis. In order to represent the competitive environment of the TRIPLE platform in the best possible way concerning different platform types and geographical origins, further 16 platforms were included in the analysis. Our selection criteria were [1] popularity, [2] diversity of platform types and [3] geographical distribution, we then came from 47 to 27 analysed platforms. Since Microsoft academic is another "academic search engine" in the USA, there were too many from the same type and same geographical region.

Point 4: Query interface. The absence of information on the inquiry form or on the functions and design of the inquiry interface stands out. If the product is aimed at researchers, it must incorporate an advanced search with complete search functions, including Boolean operators, search by fields, etc.

Response 4: Regarding the absence of information on the inquiry form, lack of functions and design of the inquiry interface we would like to specify that the design phase of the project has recently started and co-design workshops are planned for spring 2021. The final design of the main pages of the interface, including the query interface, will be then finished only in September 2021. We deal here with an iterative process and the design won't be visible before December 2021. This iterative process includes besides a user activity system relying on metrics and analytics to check what deserves improvement regarding users’ needs. Two sessions of users testing (10-15 users maximum per session) will experience quantitative & qualitative research. A milestone will report the conclusions of the testing phase and will present the final co-design of the interface. This is the reason why we did not include this task in the paper as no tangible results are now available although the search engine will contain complete search functions including boolean operators and search by fields. These features you mentioned constitute a minima and our ambition is to go further to stick more closely than usual to users requirements. This is also the reason why it did not seem to us relevant to include this state of affairs, which is in no way innovative.

Point 5: Taking the above into account, it seems like the paper is being used to publicize the TRIPLE platform. Additionally, the document does not address the question of how much citizens, such as Mr. David Green or Carolina Weber, would have to pay to access the TRIPLE platform, or who should be responsible for paying for such access.

Response 5: Concerning pricing of the platform, there is nothing set in stone. We are still in the ideation phase of our business model. For the time being we do not see a big potential at the beginning (launch of platform) to charge users for using the ordinary discovery functions. Since we do not have yet an overview on how users accept and demand the functions and features of GOTRIPLE it is hard to come up with prices and charging. But of course, we could imagine charging users for premium features but this is a bit too early to fix this. A working group is dedicated to the sustainability and business model of the model and is in charge of delivering tangible results over the next year.

Round 2

Reviewer 2 Report

I have no new comment for authors 

Author Response

Dear reviewer,

you will find here attached the new version of our paper taking into account your last remarks. In addition, please find here additional comments : 

  1. the title is the same than in the previous version. Response : indeed, the suggestion was not edited in the word document, it is corrected.
  2. In the keywords section the “community-driven approach” idea is maintained, but nothing appears in this section about the user-centric process (the supposed main idea).  Response : it is an oversight on our side, we have corrected it to replace the notion of driven community by user centric in keywords.
  3. Additionally, the idea of a community-driven approach appears in the manuscript in the same terms as in the previous version (see for example pages 2, 4, 23 or 28).    Response: here again user-centric has replaced this initial notion of community-driven all along the document.
  4. You are presenting your platform as a tool operating in the context of the European Open Science, that is, in the context of open (and free) access for every citizen. Any business model should specify the financing sources beforehand.  Your explanations (e.g. “we are still in the ideation phase of our business model”)  about this topic is surprising. It is even more surprising when you present yourself as “...a consortium, composed of 19 partners with different expertise and competences, with complementary skills and with different approaches, working together towards a common objective, is the community that drives the design and development of the GOTRIPLE platform”.Response : In the proposal we did not specify the financing sources but we have precised that one Work Package is dedicated to the exploitation and sustainability of the platform. One of its task is linked to another Work Package dedicated to co-design and user-requirements. And always in the same user centric approach, we planned to have workshops in the WP co-design and user-requirements to study at what extent we can provide a sustainable platform with an efficient business model respecting also Open Access principles. We confirm to you we have expertise in this field within the consortium but did not want to establish something definitive before having studied the market and the opportunities raised by analysis of surveys, interviews and workshops.

We hope those comments answer to your questions and eventual doubts. 

Best Regards,